# Selective ring expansion and C−H functionalization of azulenes

Sangjune Park [1,2], Cheol-Eui Kim[1,2], Jinhoon Jeong[3,4], Ho Ryu [3,4], Chanyoung Maeng[1,2], Dongwook Kim [3,4], Mu-Hyun Baik [3,4] & Phil Ho Lee [1,2,5] ✉

We report a transition metal-catalyzed ring expansion of azulene that can be contrasted with C−H functionalization. This study represents the first example of the successful ring expansion of azulene using Cu(hfacac)$_2$ (hfacac: hexafluoroacetylacetonate) with a diazo reagent. This result is notable for extending the Buchner reaction, previously limited to benzenoid aromatics, to nonbenzenoid compounds. The chemoselectivity of the reaction can be directed towards C−H functionalization by substituting the Cu catalyst with AgOTf. This approach does not require the addition of phosphine, NHC, or related ligands, and prefunctionalization of azulenes is unnecessary. Furthermore, the method exhibits excellent functional group tolerance, allowing for the synthesis of a wide range of 6,7-bicyclic compounds and C−H functionalized azulenes. We also present a theoretical study that explains the experimental observations, explaining why copper afford the ring expansion product while silver forms the C−H alkylation product.

Fused carbocycles are key structural elements of molecules in nature and they are often found in drugs and organic materials. Many synthetic methods for assembling these privileged structures have been developed, but bicyclic systems containing six and seven-membered rings proved exceedingly difficult to prepare. We found a solution by functionalizing an azulene skeleton that consists of fused five and seven-membered rings and carry out a ring expansion reaction to afford the desired bicycle. The azulene derivatives are important targets on their own right and have attracted much attention in the past. The π-electron polarization in azulene is special, because the lack of symmetry in how the two rings are fused leads to a negative charge polarization in the five-membered ring and a positive charge polarization in the seven-membered ring, giving rise to a surprisingly large dipole moment. This unusual polarization results in distinctive electronic properties and an intensive deep blue color. Azulenes found applications in material science involving optoelectronic[1] and electrochromic devices[2], nonlinear optics[3], organic electroluminescent devices[4], near-infrared quencher[5], to only name a few. Azulene derivatives as a structural motif are also used for anti-inflammatory[6] and anticancer[7] agents in medicinal chemistry.

The large dipole moment of azulenes due to their π-electron polarization is associated with their high reactivity, rendering the C1- and C3-positions of five-membered ring nucleophilic, while C4-, C6-, and C8-positions are electrophilic. Thus, electrophilic substitutions usually occur at the C1- and C3-positions, whereas nucleophilic additions are common for the C4-, C6-, and C8-positions, and C−H functionalization of azulenes are well developed for the synthesis of azulene-containing compounds (Fig. 1A)[8–18].

Diazo compounds in combination with transition metal catalysts offer a rich foundation for C−H activation[16–24], where highly reactive metal carbenoid complexes are formed by extruding dinitrogen. We envisioned that the C−H bonds of the negatively polarized five-membered ring in azulene should be reactive towards an electrophilic metal carbenoid complex. Indeed, AgOTf can effectively catalyze the

[1]Department of Chemistry, Kangwon National University, Chuncheon 24341, Republic of Korea. [2]National Creative Research Initiative Center for Catalytic Organic Reactions, Chuncheon 24341, Republic of Korea. [3]Department of Chemistry, Korea Advanced Institute of Science and Technology (KAIST), Daejeon 34141, Republic of Korea. [4]Center for Catalytic Hydrocarbon Functionalizations, Institute for Basic Science (IBS), Daejeon 34141, Republic of Korea. [5]Institute for Molecular Science and Fusion Technology, Kangwon National University, Chuncheon 24341, Republic of Korea. ✉e-mail: mbaik2805@kaist.ac.kr; phlee@kangwon.ac.kr

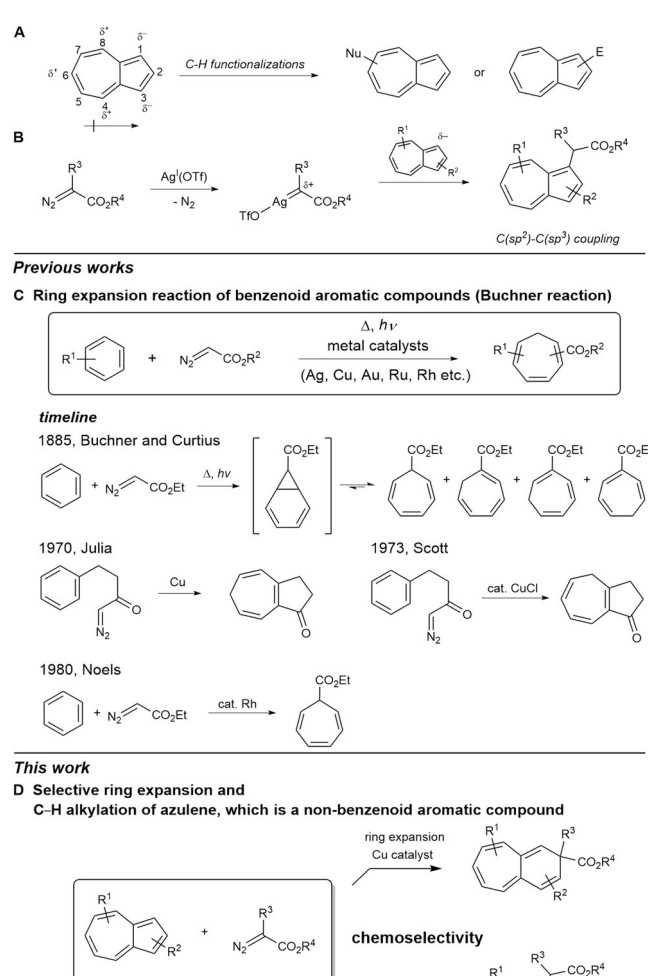

**Fig. 1 | Transformations of Azulenes. A** Dipolar properties of azulenes and general azulene functionalizations. **B** Silver(I)-catalyzed C($sp^2$)−C($sp^3$) coupling of azulenes in this work. **C** Ring expansion reaction of benzenoid aromatic compounds (Buchner reaction). **D** Selective ring expansion and C−H alkylation of azulene, which is non-benzenoid aromatic compound.

C−H functionalization of azulene in the presence of diazo compounds (Fig. 1B). Compared to the previously reported methods of C−H functionalization of azulenes[8], the silver catalyst provides direct C−H alkylation with high levels of functional group tolerance and efficiency.

While C−H activation chemistry mediated by metal carbenes is a well-known outcome, a less explored alternative reaction is the access to cyclohexadiene intermediates that can lead to ring expansion. The Buchner ring expansion reaction, reported by E. Buchner and T. Curtius in 1885, historically provided access to seven-membered rings via dearomatization of benzenoid aromatic substrates (Fig. 1C)[25]. The first step of the Buchner ring expansion reaction involves the formation of a carbene from ethyl diazoacetate, which then cyclopropanates a benzenoid aromatic ring through (2 + 1) cycloaddition. The ring expansion occurs in the second step with an electrocyclic opening of the cyclohexadiene ring to form the seven-membered ring. Since its discovery, non-catalyzed and metal-mediated variants of this reaction have become a standard method for preparing seven-membered rings from benzenoid aromatic substrates.

Conventional methods of cyclopropanation using diazo reagents have significant drawbacks. The first cyclopropanation of α-diazoketones was reported in 1970, offering a useful synthetic method for 5,7-fused bicycles, but yields were modest, and harsh

conditions were required[26]. Later, CuCl was shown to catalyze the intramolecular Buchner reaction of 1-diazo-4-phenylbutan-2-one to afford cross-conjugated dihydroazulenone[27]. However, high reaction temperatures prevented the access of cycloheptatriene, a fairly labile kinetic product. Dirhodium catalysts, pioneered by Noels[28] offered significant advances, and various substituted benzene derivatives could be used to obtain useful products with good yields and regioselectivity. Nevertheless, the Buchner reaction produced ring expansion of 6- to 7-membered rings in all cases, and the expansion of a non-benzenoid 5-membered ring to produce a 6-membered ring remained unknown to date[29]. Conversely, 3-halopyridines were prepared from the reaction of pyrrole with haloform-derived carbenes in 1881[30]. Recently, Levin and co-workers reported a reaction that selectively generates 3-arylpyridine and quinoline motifs by inserting aryl carbonyl cation equivalents into pyrrole and indole cores, respectively[31].

Herein, we present a copper-catalyzed ring expansion of non-benzenoid aromatic azulene substrates that allows the conversion of 5-membered rings to 6-membered rings under mild conditions using copper catalysts (Fig. 1D). The functional group tolerance is excellent and allows the synthesis of a wide range of 6,7-bicyclic compounds. This approach does not demand any specific ligand and prefunctionalization of azulenes. When silver catalysts are employed, the reaction can effectively be redirected towards C−H functionalization.

## Results and discussion
### Reaction optimization
Table 1 summarizes the catalytic conditions that were explored using a model reaction of azulene (**1a**) with methyl 2-diazo-2-phenylacetate (**2a**). Although copper(II) chloride and bromide were ineffective, copper(I) chloride and bromide afforded the desired 6,7-bicyclic product (**3a**) in DCE as red solids in 36% and 45% yields, respectively, together with C−H alkylation product (**4a**) (entries 1−4). Other copper(II) catalysts such as Cu(OAc)$_2$, Cu(OAc)$_2 \cdot$H$_2$O, and Cu(OTf)$_2$ produced **3a** in dioxane in yields ranging from 32% to 39% (entries 5−7). Further screening of catalysts, including Cu(acac)$_2$, Cu(tfacac)$_2$, and Cu(hfacac)$_2$, (acac: acetylacetone; tfacac: trifluoroacetylacetone; hfacac: hexafluoroacetylacetone) revealed that Cu(hfacac)$_2$ is an optimal catalyst to selectively furnish **3a** in 81% yield (entries 8−10). Among the solvents tested, dichloroethane (DCE) gave the best results, although dioxane, hexane, toluene, chloroform, and acetonitrile were also effective (entries 10−15). Dilution of the reaction mixture from 0.1 M to 0.05 M increased the yield and the best result was achieved through the treatment of **1a** (0.2 mmol, 0.1 equiv) with **2a** (1.0 equiv) in the presence of Cu(hfacac)$_2$ (2.0 mol %) in DCE at 25 °C for 30 min, which resulted in the 6,7-bicyclic product (**3a**) in 95% yield (entry 17). In contrast, silver(I) triflate (2.0 mol %) was an effective catalyst for the C−H alkylation of **1a** with **2a**, affording selectively **4a** as a blue solid in 74% yield (entry 19). In addition, a variety of diazo compounds, including ethyl diazoacetate, ethyl vinyl diazoacetate, α-diazo-β-keto ester, diazomalonate, and α-diazo oxime ether, were examined (see the Supplementary Table S2). Ethyl diazoacetate and ethyl vinyl diazoacetate was not effective. Although reaction of azulene with diazomalonate provided the ring expansion and C−H alkylation product (39% and 25%, respectively), α-diazo-β-keto ester and α-diazo oxime ether were selectively converted to the C−H alkylation product (53%) and the ring expansion product (83%). These results indicate that the structure of diazo compound is critically important for successful reactions. Other reaction parameters were tested, as discussed in the Supplementary Information.

### Substrate scope
With the optimized conditions in hand, the scope and limitation of diazoesters and azulenes were scrutinized (Fig. 2). Variation of

## Table 1 | Reaction optimization[a]

| Entry | Cat. (mol %) | Solvent | Temp. (°C) | Conv. (%) | Yield (%)[b] | |
|---|---|---|---|---|---|---|
| | | | | | 3a | 4a |
| 1 | CuCl | DCE | 25 | 100 | 36 | 7 |
| 2 | CuBr | DCE | 25 | 80 | 45 | 9 |
| 3 | CuCl$_2$ | DCE | 70 | 100 | 0 | 0 |
| 4 | CuBr$_2$ | DCE | 70 | 100 | 0 | 0 |
| 5 | Cu(OAc)$_2$ | dioxane | 70 | 70 | 39 | 7 |
| 6 | Cu(OAc)$_2$·H$_2$O | dioxane | 70 | 65 | 38 | 9 |
| 7 | Cu(OTf)$_2$ | dioxane | 25 | 100 | 32 | 0 |
| 8 | Cu(acac)$_2$ | dioxane | 40 | 62 | 55 | 2 |
| 9 | Cu(tfacac)$_2$ | dioxane | 40 | 78 | 42 | 1 |
| 10 | Cu(hfacac)$_2$ | dioxane | 40 | 87 | 81 (78)[c] | 0 |
| 11 | Cu(hfacac)$_2$ | hexane | 25 | 84 | 58 | 1 |
| 12 | Cu(hfacac)$_2$ | toluene | 25 | 100 | 64 | 1 |
| 13 | Cu(hfacac)$_2$ | DCE | 25 | 100 | 84 (82)[c] | 0 |
| 14 | Cu(hfacac)$_2$ | CHCl$_3$ | 25 | 98 | 68 | 2 |
| 15 | Cu(hfacac)$_2$ | MeCN | 40 | 85 | 53 | 4 |
| 16 | Cu(hfacac)$_2$ | DMF | 40 | 30 | 6 | 0 |
| 17[d] | Cu(hfacac)$_2$ | DCE | 25 | 100 | 95 (93)[c] | 0 |
| 18[d,e] | Cu(hfacac)$_2$ | DCE | 25 | 100 | 90 | 0 |
| 19[d,f] | AgOTf | DCE | 25 | 100 | 0 | 74[c] (5)[c,g] |

[a]Azulene (**1a**, 0.2 mmol, 1.0 equiv), diazo compound (**2a**, 1.0 equiv), and catalyst (2.0 mol %) were used in solvent (1.0 mL, 0.1 M) under a N$_2$ atmosphere.
[b]NMR yield with CH$_2$Br$_2$ as an internal standard.
[c]Isolated yield.
[d]DCE (4.0 mL, 0.05 M) was used.
[e]Catalyst (1.0 mol %) was used.
[f]**1a** (2.0 equiv) and **2a** (0.2 mmol, 1.0 equiv) were used.
[g]1,3-Dialkylated product.

substituents on the aryl group of aryl diazoesters **2** was investigated and found to have a little effect on the efficiency of the ring expansion reaction. Products using aryl diazoesters (**2**) bearing electron-donating substituents such as methyl (**2b**–**2d**) and methoxy (**2e** and **2f**) on the aryl ring were formed in good yields ranging from 70% to 82% at 25 °C. When methylene-3,4-dioxy-substituted phenyl diazoester was exposed to azulene **1a** in the presence of 2.0 mol % Cu(hfacac)$_2$, the expected 6,7-fused bicyclic ring expansion product (**3g**) was obtained in 91% yield at 25 °C in 30 min with the release of a dinitrogen molecule. Aryl diazoesters (**2**) having electron-withdrawing groups such as chloro-, bromo-, ester-, and nitrile group are applicable to the present transformation, affording the corresponding ring expanded compounds (**3h**–**3k**) in good to excellent yields. Notably, aryl diazoesters (**2**) bearing labile trimethylsilyl-, pinacolboryl-, azo, and activated olefinic groups underwent the ring expansion reaction with azulene efficiently at 25 °C. The structure of **3o** was confirmed by X-ray crystallography. When ethyl diazoester (**2p**) possessing an estrone moiety was employed as the substrate, the ring expansion product **3p** was obtained in 51% yield. Alkyl diazoesters were compatible with the reaction conditions. For example, methyl and phenethyl diazoesters were applied to the present method, providing the desired ring expansion product **3q** and **3r** in 84% and 87% yields, respectively.

Next, the Cu-catalyzed ring expansion reaction was tested for a wide range of azulenes. The substituents on the cyclopentadienyl ring of the azulene have little influence on the reaction efficiency. The ring expansion reaction was amenable to azulenes having substituents such as phenyl-, 4-methylphenyl-, 3-chlorophenyl-, and 4-trifluoromethylphenyl groups at the C1 position to deliver the desired products **3s**–**3v** in good to excellent yields in the range of 70–93%. Interestingly, the presence of potentially reactive olefinic double bonds on the aryl ring did not deteriorate the selectivity between ring expansion reaction of the cyclopentadienyl ring. The presence of tosylamino group in azulene did not influence the outcome of the ring expansion reaction, thus affording **3w** in 88% yield. The ring expansion reaction occurred exclusively on the cyclopentadienyl ring of azulene even in the presence of electron-rich 1-phenyl-2-(N-tosylamino)ethenyl group. Also, azulenolactone underwent the ring expansion reaction. Guaiazulene having methyl as well as isopropyl group on the cycloheptatrienyl ring of azulene was readily ring-expanded with phenyl-, 4-methylphenyl-, and 4-bromophenyl diazoesters, producing the desired ring expanded 6,7-fused bicyclic compounds (**3z**, **3a'**, and **3b'**) in moderate to good yields ranging from 68% to 85%. When 4,6,8-trimethylazulene was treated with methyl phenyl diazoacetate in the presence of the Cu-catalyst in dioxane, the desired product **3c'** was obtained in 88% yield. Likewise, a ring expansion reaction using 4,6,8-trimethylazulenylmethyl phenyl diazoacetate was attempted under the standard conditions, producing the ring expanded tricyclic azulenyl compound **3d'** as an orange solid in 95% yield.

**Fig. 2 | Scope of azulenes and diazoesters in ring expansion reaction[a]. [a]1** (0.2 mmol, 1.0 equiv), **2** (1.0 equiv), and Cu(hfacac)$_2$ (2.0 mol %) were used in DCE (4.0 mL) at room temperature for 30 min under a N$_2$ atmosphere. [b]Diazo compound (1.5 equiv) was used. [c]Diastereomeric ratio. [d]Cu(hfacac)$_2$ (4.0 mol %) was used. [e]Dioxane was used as a solvent.

Next, we examined the scope and limitation of the C–H alkylation for variously substituted azulenes **1** in the reaction with diazoacetate (**2**) (Fig. 3). The reaction efficiency was not influenced by the electronic properties of the azulenes. When 1-phenylazulene was exposed to 1.0 equivalent of methyl phenyl diazoacetate in the presence of 2.0 mol % AgOTf at 25 °C, the desired methyl 2-phenyl-2-[(3-phenyl)azulen-1-yl] acetate **4b** was obtained in 80% yield with the extrusion of a dinitrogen molecule. Azulenes having substituents, including 4-methylphenyl and 3-chlorophenyl, at C1 position underwent the C–H alkylation at 25 °C to give the corresponding azulenes **4c** and **4d** in 86% and 72% yields, respectively. It was noteworthy that the selectivity between C–H alkylation on cyclo-pentadienyl ring and the Buchner reaction on the aryl ring was maintained. Benzoyl- and 1-phenyl-2-(*N*-tosylamino)ethyl-sub-stituted azulenes took part in the C–H alkylation reaction with dia-zoacetate to afford the corresponding products **4e** and **4f** in good yields. In addition, azulenolactone was subjected to the C–H alky-lation reaction. When guaiazulene was treated with methyl phenyl diazoacetate in the presence of Ag catalyst, the desired alkylated azulene **4h** was obtained in 74% yield in spite of steric congestion.

Because 1-(4-methylphenyl)azulene gave the alkylation product (**4c**) in highest yield (86%), this substrate was treated with a wide range of aryl diazoacetates to investigate the efficiency of the Ag-catalyzed C–H alkylation reaction. Electronic modification of substituents on the aryl group of alkyl aryl diazoacetates **2** was examined and had little effect on the efficiency of the C–H alkylation. The functionalized azulenes (**4i**–**4l**) were produced from the reaction of azulene with a variety of aryl diazoesters (**2**) having electron-donating substituents including methyl and methoxy on the aryl ring in good yields varying from 70% to 82% at 25 °C. Methylene-3,4-dioxy-substituted phenyl diazoester turned out to be compatible with the reaction conditions, providing the desired product **4 m** in 72% yield. Aryl diazoacetates bearing electron-withdrawing groups, including chloro-, bromo-, ester-, and nitrile on the aryl ring, are applicable to the C–H alkylation, producing the desired azulenes (**4n**–**4q**) in good to excellent yields ranging from 65% to 91%. Aryl diazoacetates bearing labile tri-methylsilyl-, pinacolboryl-, methylsulfinyl-, and activated olefinic groups were also compatible with the reaction conditions. When dia-zoacetate having an estrone moiety underwent the C–H alkylation reaction with 1-(4-methylphenyl)azulene, the corresponding product **4v** was obtained in 48% yield. No ring expansion product was observed in all cases.

## Mechanistic studies and DFT calculations
The dramatically different reactivities toward azulenes depending on the choice of metal are difficult to understand. Thus, we conducted a detailed mechanistic study using density functional theory

**Fig. 3 | Scope of azulenes and diazoesters in C−H alkylation[a].** [a]**1** (0.2 mmol, 1.0 equiv), **2** (1.5 equiv), and AgOTf (2.0 mol %) were used in DCE (4.0 mL) at 25 °C for 1 h under a N₂ atmosphere. [b]**1a** (2.0 equiv) and **2a** (0.2 mmol, 1.0 equiv) were used. 1,3-Dialkylated product was obtained in 5% yield. [c]Reaction temperature: 50 °C. [d]Methyl phenyl diazoacetate (3.0 equiv) was used. [e]Reaction temperature: 70 °C. [f]Diastereomeric ratio.

calculations using the popular B3LYP and M06 functionals (full technical details are provided in the Supplementary Information), which produced the catalytic cycles shown in Fig. 4A to explain the experimental observations. The reaction begins with the formation of a metal-carbenoid species extruding dinitrogen and the nucleophilic carbon in azulene attacks the electrophilic α-carbon in the metal-carbenoid. This azulene alkylation into the metal-carbenoid species gives a metal alkyl complex, which is energetically favorable partially due to the stabilization of a positive charge in the azulene by delocalization of the π-electrons over 10 carbons. The details of the metal alkyl complex formation are presented in the Supplementary Information. Our calculations suggest that the chemoselectivity originates from two independent reaction pathways, as illustrated in Fig. 4B. For the ring expansion, the metal alkyl complex undergoes cyclization giving a cyclopropane intermediate, which ring-expands to form a 6,7-fused bicyclic product. The C−H alkylation proceeds via a 1,4-proton shift followed by tautomerization. Details of several other possible reaction pathways that we considered are given in the Supplementary Information.

Figure 4B compares the two possible reaction pathways for the copper alkyl complex A4 and the silver alkyl complex B4. As previously reported, copper(II) complexes are easily reduced to copper(I) in the presence of diazo compounds[32–34]. In the copper catalysis, we observed darkening of the reaction mixture by the formation of copper(0) species and the homo-coupling side product of the diazo compound[32], which indicates the formation of copper(I) species. Therefore, the catalytic reaction begins with Cu[I](hfacac) as an active catalyst instead of Cu[II](hfacac)₂. The copper alkyl complex **A4** can either undergo a 1,4-proton shift, where a carbonyl oxygen in the ester group deprotonates the β-proton in the azulene moiety passing through the transition state **TS(A4-A5)**

or a cyclization via the transition state **TS(A4-A8)**. Since the transition state **TS(A4-A8)** is 3.4 kcal/mol lower in energy **TS(A4-A5)**, the reaction is predicted to dominantly form the fused cyclopropane intermediate **A8**. To force the reaction forward and promote ring expansion, a C−C bond cleavage must take place. This reaction traverses the transition state **TS(A8-A9)** with a barrier of 20.1 kcal/mol and forms the ring-expanded product **A9** at a relative solution phase free energy of −8.3 kcal/mol. Meanwhile, the C−H alkylation pathway in the copper catalysis is energetically higher than the ring expansion pathway. A 1,4-proton shift of **A4** furnishes an enol intermediate **A5**, which is downhill in energy by −6.4 kcal/mol, and two equivalents of **A5** forms the adduct **A6** followed by tautomerization via the transition state **TS(A6-A7)**. The calculated overall barrier for this reaction pathway is 22.3 kcal/mol, and is 3.1 kcal/mol higher than **TS(A8-A9)**, rendering the the C−H alkylation pathway unlikely with the copper catalyst.

Interestingly, the silver catalyzed reaction shows dramatically different barriers for these two possible reaction pathways. The silver ion is more polarizable than the copper ion and effectively stabilizes a negatively-charged alkyl-substrate in **B4**, thus the energy of ring expansion reaction in silver catalysis is up-shifted compared to the copper catalysis. The transition state **TS(B4-B8)** is found at 12.4 kcal/mol, and the cyclization affords **B8** at −3.3 kcal/mol. The C−C bond cleavage requires 18.8 kcal/mol, generating the product **B9** with a solution phase free energy of −5.1 kcal/mol. Intriguingly, the energy ordering displayed by the two metals in the C−H alkylation pathway is reversed when compared to the ring-expansion reaction. Although the enol intermediate **B5** is slightly higher in energy at −3.4 kcal/mol than the copper analogue **A5**, the silver adduct **B6** is significantly lower than **A6**, the adduct formed by copper. This is easy to understand as the silver ion can

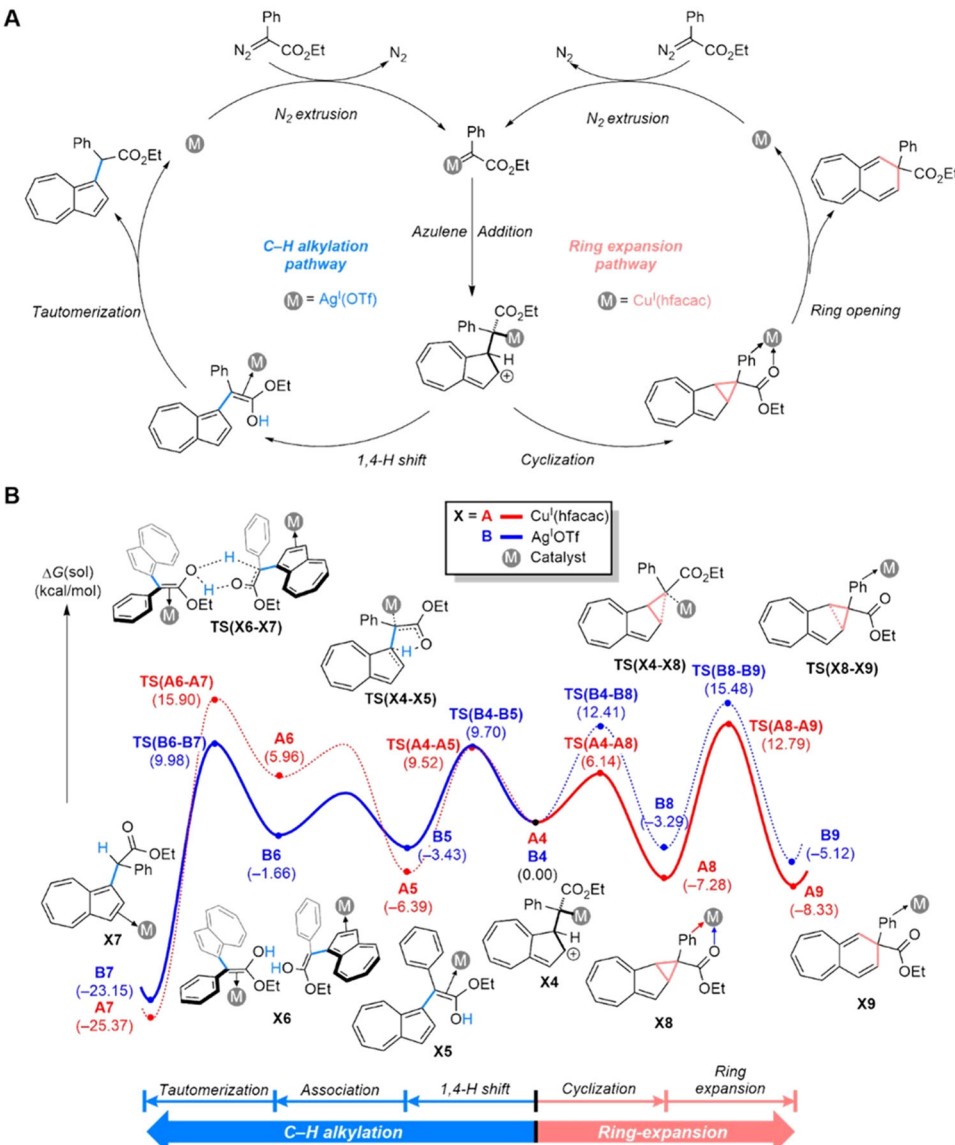

**Fig. 4 | A Proposed mechanism. A** Proposed mechanism of ring expansion and C–H alkylation pathways. **B** Computed reaction profiles of ring expansion and C–H alkylation pathways catalyzed by either Cu(hfacac)₂ or AgOTf. The dotted lines indicate a minor pathway.

accommodate higher coordination numbers and can engage π-electron donors more effectively than copper. As a result, the silver ion in **B6** displays an additional interaction with the aryl group attached to the enol functionality. This advantage of silver over copper causes the enol intermediate to interact more strongly with the metal. As a consequence, the tautomerization barrier is much lower when silver is used. The related transition state **TS(B6-B7)** is located at 10.0 kcal/mol and is 5.5 kcal/mol lower than **TS(B8-B9)**, giving rise to a dominant formation of the product **B7**.

In conclusion, we developed a transition metal-catalyzed ring expansion and C–H functionalization of azulene. The chemoselectivity can be controlled simply by changing the metal ion. A theoretical study was able to explain the experimental observations that copper affords the ring expansion product while silver results in the formation of C–H alkylation products. The α-carbon in the transition state for the cyclization is a relatively hard Lewis base. It interacts much more weakly with the soft silver ion than the copper ion, leading to the observed selectivity. This work constitutes the first example of the ring expansion of azulene using diazo

compounds, offering a stark contrast to the alternative C–H functionalization reaction. This achievement is particularly noteworthy because it extends the scope of the Buchner reaction, a transformation previously limited to benzenoid aromatics, to non-benzenoid counterparts. This method does not demand any specific ligand and prefunctionalization of azulenes. In addition, the excellent functional group tolerance allows the synthesis of a wide range of 6,7-bicyclic compounds.

## Methods

### General procedure for ring expansion of azulenes with alkyl and aryl diazo esters

Cu(hfacac)₂ (1.9 mg, 2.0 mol %), azulene derivatives (0.2 mmol), and DCE (3.0 mL) were added to a test tube equipped with a stirring bar. To the solution was added aryl diazoacetate derivatives (0.2 mmol) in DCE (4.0 mL). The mixture was stirred at 25 °C for 30 min under a nitrogen atmosphere. The residue was passed through a pad of Cellite to remove Cu(hfacac)₂ and eluted with CH₂Cl₂. The filtrate was concentrated under reduced pressure and the residue was purified by flash column chromatography (EtOAc:hexane) to give **3**.

## General procedure for C–H functionalization of azulenes with aryl diazo esters

AgOTf (1.0 mg, 2.0 mol %), azulene derivatives **1** (0.2 mmol), and DCE (3.0 mL) were added to a test tube equipped with a stirring bar. To the solution was added aryl diazoacetate derivatives **2** (0.3 mmol) in DCE (1.0 mL). The mixture was stirred at 25 °C for 1 h under a nitrogen atmosphere. The residue was passed through a pad of Cellite to remove AgOTf and eluted with $CH_2Cl_2$. The filtrate was concentrated under reduced pressure and the residue was purified by flash column chromatography to give **4**.

## Computational methods

All calculations were performed at the density functional theory (DFT) level as implemented in the Jaguar 9.1[35]. Geometry optimizations were carried out with the Becke's three-parameter exchange functional (B3LYP)[36, 37] with D3 correction[38] and 6–31 G** basis set[39]. Copper and silver were represented using the Los Alamos double zeta basis, which contains effective core potentials[40–42]. The energies of the optimized structures were reevaluated with a triple-ζ basis set cc-pVTZ(-f)[43] for the main group elemental and with LACV3P for the copper and silver, using the Minnesota functional M06[44], as this protocol produced reaction energies that were most consistent with the experimental observations. Vibrational frequency calculations were performed using the same level of theory used for geometry optimization. The zero-point energy (ZPE) values and entropy were obtained from the frequency calculations. Solvation energies were simulated by a self-consistent reaction field (SCRF)[45–47] approach with the dielectric constant ε = 10.125 (DCE) using the optimized structures in the gas phase.

The energy components have been computed with the following protocol. The free energy in solution phase G(sol) has been calculated as follows:

$$G(sol) = G(gas) + G(solv) \quad (1)$$

$$G(gas) = H(gas) - TS(gas) \quad (2)$$

$$H(gas) = E(SCF) + ZPE \quad (3)$$

$$\Delta E(SCF) = \sum E(SCF) \text{for products} - \sum E(SCF) \text{for reactants} \quad (4)$$

$$\Delta G(sol) = \sum G(sol) \text{for products} - \sum G(sol) \text{for reactants} \quad (5)$$

G(gas) is the free energy in gas phase; G(solv) is the free energy of solvation; H(gas) is the enthalpy in gas phase; T is the temperature (298.15 K); S(gas) is the entropy in gas phase; E(SCF) is electronic energy as computed from the SCF procedure and ZPE is the zero point energy. The entropy we refer involves vibrational, rotational, and translational entropy of the solute(s) and the entropy of the solvent is implicitly included in the continuum solvation model.

## Data availability

The data supporting the findings of this study are available within this article and its Supplementary Information, which contains experimental details, characterization data, copies of NMR spectra for all new compounds, and DFT calculation data. Crystallographic data for **3o**, **3y**, and **4g** have been deposited at the Cambridge Crystallographic Data Centre (CCDC) under deposition numbers CCDC 1435091, 1916356, and 1916358, respectively. Copies of the data can be accessed free of charge via https://www.ccdc.cam.ac.uk/structures/. All other data are available from the corresponding author upon request. Source data are provided with this paper.

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

## Acknowledgements

This work was supported by the National Research Foundation of Korea (NRF) grant funded by the Korea government (MSIP) (2021R1A2C3008862 and RS-2023-00271205). This work was also supported by the Institute for Basic Science (IBS-R010-A1) in Korea.

## Author contributions

P.H.L. and M.-H.B. conceived and designed the project and wrote the manuscript. S.P., C.-E.K., and C.M. carried out the experiments. J.J. and H.R. performed computational studies. D.K. performed the single-crystal X-ray diffraction analysis. P.H.L. organized the research and all authors analyzed the data, discussed the results, and commented on the manuscript.

## Competing interests

The authors declare no competing interests.
