## [Peer Review File · Nature Communications]

REVIEWER COMMENTS

Reviewer #1 (Remarks to the Author):

This study describes two distinct reactions between azulenes and aryldiazoacetates, which is dependent on which catalyst is used. In the case of copper-catalyzed reactions, a ring expanded product is generated, while in the case of silver catalyzed reactions, alkylation products are generated. The alkylated product has been several times previously but as far as I am aware the indole ring expanded common has not been reported previously. A good systematic study has been carried out showing either type of products can be effectively generated and a follow-up computational study nicely explains the cause of the catalyst influence.

The study is sound and of high quality. One area that needs to be improved is to place the literature precedence correctly. Scheme 1 suggested that the reaction of carbenes with azulenes is new but in actual fact there are many examples of the metal-catalyzed carbene alkylation of azulenes, including the use of diazoacetates, aryldiazoacetates and sulfonyltriazoles as the carbene sources. The introduction should give clear recognition of the previous azulene alkylation work.

It would be of interest to know what is the outcome of dirhodium catalyzed reactions under the optimized conditions. There are many examples of rhodium-catalyzed reactions of aryldiazoacetates with electron rich alkenes or dienes in which cyclopropanes are formed when hydrocarbon solvents are used but products derived from zwitterionic intermediates are formed when dichloromethane is used. Does the same product switch seen here occur under dirhodium catalysis by simply changing the solvent?

The hard question is whether this work is sufficiently innovative for Nature Communications? The general outcome is quite common for reactions of metal carbenes with electronic rich alkenes - the intermediacy of zwitterionic structures that can undergo either cyclopropanation or other side products. So, the formation of the two types of products is not that surprising although it is elegant that the product outcome can be controlled by using the appropriate catalyst. Granted the authors have developed improved conditions for the alkylation but the innovative component of this work is the copper-catalyzed ring expansion. Overall, I am leaning towards publication, primarily because the work is of high quality and the study is clear and concise.

Reviewer #2 (Remarks to the Author):

The paper described the reaction between diazo compounds and azulenes, in which Cu catalysts favored the ring expansion and Ag catalysts favored the C–H insertion. While the different results with different catalysts are mechanistically interesting, these two reaction modes are widely known in simple benzenes. The researchers only expanded the scope from normal aromatic systems to azulenes. Therefore, the novelty is moderate. Besides, the authors attempted to explain the different reactivities via DFT calculation. While the DFT data fully matched the experimental results, the explanation is not convincing and the ‘energy decomposition analysis’ part is very confusing and possibly incorrect.

1) Since this is a ring expansion reaction from 5-membered rings to 6-membered rings with carbene intermediates, references about the Ciamician-Dennstedt reaction should be included and discussed in the introduction part, for example, the initial report *Ber. Dtsch. Chem. Ges.* 1881, 14, 1153 and some recent work *J. Am. Chem. Soc.* 2021, 143, 11337. Mechanistically speaking, the reaction itself is more like the Ciamician-Dennstedt reaction. They both include electron-rich 5-membered rings.

2) Some uncommon abbreviations require detailed explanations, for example, Cu(tfacac)₂ and Cu(hfacac)₂.

3) I am wondering if diazo compounds with different structures could be tolerated, for example, α -diazo malonic esters or α -diazoketones.

4) Since the authors proposed that the active catalyst is Cu(I) instead of Cu(II), it’s better to add some Cu(I) catalyst entries in the screening table.

5) In page 11 line 224, enolate is probably not ‘a hard Lewis base’ and this theory may not give convincing explanations on the reactivity difference. A detailed discussion is suggested.

6) The authors claimed that they conducted ‘energy decomposition analysis’ on the key transition states. In both the manuscript and SI, we didn’t find any reference or procedures about the technique details of the ‘energy decomposition analysis’. According to our understanding, the energy decomposition analysis is to separate an interaction into three parts: electrostatic interaction, repulsive exchange interaction and orbital interaction (see: *WIREs Comput Mol Sci*, 2018, 8: e1345. doi: 10.1002/wcms.1345). What the authors did is more like a ‘distortion-interaction analysis’ (see: *ACIE*, 2017, 56, 10070). However, what the authors showed in the SI is still very confusing. In Fig. S4, the interaction should decrease the overall barrier instead of increasing it. We strongly suggest to include more details or references for better demonstration.

7) The authors proposed that both C–H insertion and ring expansion pathways went through the initial addition step with a common intermediate A4 or B4. They said in the SI that the concerted C–H insertion transition states could not be located, but no discussion about the concerted cyclopropanation pathway was included. I am wondering if a concerted cyclopropanation pathway without the insertion intermediate is possible.

8) ^{13}C spectra of 1v and 3v are misinterpreted. At least three quartet should be observed due to fluorine coupling with coupling constant values of ~ 270 Hz (1J), ~ 30 Hz (2J), ~ 4 Hz (3J). We suggest more scans on a more concentrated sample for better spectra.

Overall, this is a borderline case. In its current form, this paper is below the standards of Nat. Commun.

Reviewer #3 (Remarks to the Author):

In this manuscript, Lee et al. have established the first example of ring expansion and C–H functionalization of azulene with diazo compound catalyzed by transition-metal complexes. The chemoselectivity of the reaction could be controlled simply by changing different transition-metal catalysts. The ring expansion of azulene could be realized by using copper catalyst and the C–H functionalization of azulene could be accomplished using silver catalyst. This strategy has excellent chemoselectivity and functional group tolerance, which could allow the synthesis of a wide range of 6,7-bicyclic compounds and C–H functionalized azulenes. Meanwhile, the reaction mechanism of ring expansion and the C–H functionalization of azulene were studied using DFT method, and the difference in the chemoselectivity of the reaction catalyzed by different metal catalysts was explained. However, the authors have to revise their manuscript to fully address the following issues before a final decision is reached.

1. The authors states that the catalytic reaction begins with $\text{CuI}(\text{hfacac})$ as an active catalyst instead of $\text{CuII}(\text{hfacac})_2$, which our calculations show to be ineffective. However, I don't find the related calculations in the manuscript or the supporting materials. Please supply this part or prove the experiment by related copper catalysts, such as $\text{CuI}(\text{hfacac})$.
2. In the section of Mechanistic studies and DFT calculations, the ring expansion reaction and the C–H functionalization catalyzed by silver catalyst and copper catalyst were calculated, respectively. There is no doubt about the reaction mechanism of azulene catalyzed by silver catalyst, but this is not the case with copper catalyst. Therefore, the authors should discuss not only the difference between $\text{TS}(\text{X4-X5})$ and $\text{TS}(\text{X4-X8})$ but also the difference in the following fundamental steps in two reaction modes. For the reaction by Ag and Cu catalyst, the authors could use the energetic span model to explain the origin of different chemoselectivity.
3. In the manuscript, the authors mentioned that the process from A8 to C1 and the process from C1 to C2 are irreversible. Why are they irreversible?
4. In the ring-expansion pathway, is it possible that Cu species didn't dissociate from O to coordinate the EtO-C(=O) moiety? This might make the ring-expansion easier.

5. It is not right to plot a wavy line connecting X5 and X6 in the association step. There is no transition states between them.

6. The format of G in Fig. 4 should be italic. In addition, the calculated data should keep one decimal place instead of two decimal places, which is beyond the error range of DFT.

Reviewer #1 (Remarks to the Author):

This study describes two distinct reactions between azulenes and aryldiazoacetates, which is dependent on which catalyst is used. In the case of copper-catalyzed reactions, a ring expanded product is generated, while in the case of silver catalyzed reactions, alkylation products are generated. The alkylated product has been several times previously but as far as I am aware the indole ring expanded common has not been reported previously. A good systematic study has been carried out showing either type of products can be effectively generated and a follow-up computational study nicely explains the cause of the catalyst influence.

The study is sound and of high quality. One area that needs to be improved is to place the literature precedence correctly. Scheme 1 suggested that the reaction of carbenes with azulenes is new but in actual fact there are many examples of the metal-catalyzed carbene alkylation of azulenes, including the use of diazoacetates, aryldiazoacetates and sulfonyltriazoles as the carbene sources. The introduction should give clear recognition of the previous azulene alkylation work.

Reply: Thank you for the feedback on our manuscript. We agree that the alkylation of azulene is a mature field with many excellent research contributions. In the introduction, we aimed to acknowledge some of this work and specifically mentioned the diazo chemistry and the use of sulfonyltriazoles (ref. 11), which the referee highlighted. As the referee points out correctly, the Ag-catalyzed alkylation we describe in this work is an improvement over previous alkylation methods, whereas the ring expansion is new. We agree with the reviewer that this point was not made clearly enough in our previous version of the manuscript. We have edited the introduction to make this point more clearly. We are a bit reluctant, however, to use more space in the intro to review previous efforts on C–H activation, as the ring expansion is more important for the core message of the paper. We have edited the intro and Figure 1 to accommodate this reviewer's concerns.

It would be of interest to know what is the outcome of dirhodium catalyzed reactions under the optimized conditions. There are many examples of rhodium-catalyzed reactions of aryldiazoacetates with electron rich alkenes or dienes in which cyclopropanes are formed when hydrocarbon solvents are used but products derived from zwitterionic intermediates are formed when dichloromethane is used. Does the same product switch seen here occur under dirhodium catalysis by simply changing the solvent?

Reply: Rh-catalyzed reaction of azulene (**1a**) with methyl 2-diazo-2-phenylacetate (**2a**) was investigated (Table S1). When **1a** was reacted with **2a** in the presence of a number of rhodium(II) catalysts (2.0 mol % each), such as Rh₂(OAc)₄, Rh₂(Oct)₄, Rh₂(TFA)₄, and Rh₂(esp)₂, in dichloroethane (DCE) at 25 °C for 5 h, poor results were obtained compared to Cu or Ag catalyst in terms of yield and selectivity (entries 1-4). Also, when a variety of solvents, such as DCE, dichloromethane (DCM), acetonitrile, toluene, *n*-hexane, and THF, were screened in the presence of Rh₂(OAc)₄ catalyst (2.0 mol %), inferior results were obtained compared to Cu or Ag catalyst in terms of yield and selectivity (entries 1 and 5-9). When DCE and DCM in the presence of Rh₂(OAc)₄ (2.0 mol %) were used as solvents, both ring expansion and alkylation products were obtained in 56% (**3a:4a** = 1:2.5) and 61% (**3a:4a** = 1:1.1) yields, respectively (entries 1 and 5). For hexane, the products were obtained in 27% (**3a:4a** = 1.7:1.0) yield together with dialkylation product (**4a'**) in 3% yield (entry 8). Although the product switch (1:2.5 → 1.7:1) by simply changing the solvent occurred, the yield was not good. In the end, when AgOTf and Cu(hfacac)₂ catalysts (2.0 mol % each) were used in the reaction of **1a** with **2a**, yield and selectivity were the best. Table S1 was added to the Supporting Information.

Table S1. Examination of Dirhodium Catalyst and Solvent in the Reaction of Azulene with Methyl Diazo Phenylacetate^a

Entry	Catalyst (2.0 mol %)	Solvent	Yield (%) ^b		
			3a	4a	4a'
1	Rh ₂ (OAc) ₄	DCE	16	40	0
2	Rh ₂ (Oct) ₄	DCE	15	22	0
3	Rh ₂ (TFA) ₄	DCE	0	25	6
4	Rh ₂ (esp) ₂	DCE	0	9	0
5	Rh ₂ (OAc) ₄	DCM	29	32	0
6	Rh ₂ (OAc) ₄	MeCN	30	7	0
7	Rh ₂ (OAc) ₄	toluene	21	33	2
8	Rh ₂ (OAc) ₄	hexane	17	10	3
9	Rh ₂ (OAc) ₄	THF	14	11	0

^a**1a** (0.2 mmol, 1.0 equiv), **2a** (1.0 equiv), and Rh catalyst (2.0 mol %) were used in solvent (0.05 M) at 25 °C for 5 h under a N₂ atmosphere. ^bNMR yield with CH₂Br₂ as an internal standard.

The hard question is whether this work is sufficiently innovative for Nature Communications? The general outcome is quite common for reactions of metal carbenes with electronic rich alkenes - the intermediacy of zwitterionic structures that can undergo ether cyclopropanation or other side products. So, the formation of the two types of products is not that surprising although it is elegant that the product outcome can be controlled by using the appropriate catalyst. Granted the authors have developed improved conditions for the alkylation but the innovative component of this work is the copper-catalyzed ring expansion. Overall, I am leaning towards publication, primarily because the work is of high quality and the study is clear and concise.

Reply: We thank the referee for recognizing that our work is novel and of high quality. We would like to highlight that we offer a very compelling and conceptually simple rational for the somewhat puzzling observation of the chemoselectivity just based on the nature of the metal ion. We feel that this is the strongest point of the paper and we have attempted to make this more prominent in the revised manuscript.

Reviewer #2 (Remarks to the Author):

The paper described the reaction between diazo compounds and azulenes, in which Cu catalysts favored the ring expansion and Ag catalysts favored the C–H insertion. While the different results with different catalysts are mechanistically interesting, these two reaction modes are widely known in simple benzenes. The researchers only expanded the scope from normal aromatic systems to azulenes. Therefore, the novelty is moderate. Besides, the authors attempted to explain the different reactivities via DFT calculation. While the DFT data fully matched the experimental results, the explanation is not convincing and the ‘energy decomposition analysis’ part is very confusing and possibly incorrect.

Reply: We appreciate the reviewer's feedback on the novelty of our work. While we respect his/her opinion, we respectfully disagree that our method is not novel. We provide a general and previously unknown method for yielding the (6,7) ring product, which has not been reported before in the literature. We understand that alkylations and ring-expansions are established for benzene substrates, but our ring-expansion affording the (6,7) ring product is surprising and novel, especially considering that ring-expansion for azulenes has not been reported previously. We have edited the introduction to emphasize this point.

Regarding the DFT results, we apologize for any confusion caused by our presentation. Our intention was to describe the computed energy profile and explain the chemoselectivity using our DFT results. We are confident that our findings are correct, but we understand that our presentation may have been unclear. We have made efforts to improve our presentation and address the referee's concerns.

1) Since this is a ring expansion reaction from 5-membered rings to 6-membered rings with carbene intermediates, references about the Ciamician-Dennstedt reaction should be included and discussed in the introduction part, for example, the initial report *Ber. Dtsch. Chem. Ges.* **1881**, *14*, 1153 and some recent work *J. Am. Chem. Soc.* **2021**, *143*, 11337. Mechanistically speaking, the reaction itself is more like the Ciamician-Dennstedt reaction. They both include electron-rich 5-membered rings.

Reply: We have added the references mentioned by the referee, as shown below, in the main text.

On the other hand, 3-halopyridines were prepared from the reaction of pyrrole with haloform-derived carbenes in 1881²⁷. Recently, Levin and co-workers reported a reaction that selectively generates 3-arylpyridine and quinoline motifs by inserting aryl carbonyl cation equivalents into pyrrole and indole cores, respectively²⁸.

27. Ciamician, G. L., Dennstedt, M. Ueber Die Einwirkung Des Chloroforms Auf Die Kaliumverbindung Pyrrols. *Ber. Dtsch. Chem. Ges.* **14**, 1153-1163 (1881).

28. Dherange, B. D., Kelly, P. Q., Liles, J. P., Sigman, M. S., Levin, M. D. Carbon Atom Insertion into Pyrroles and Indoles Promoted by Chlorodiazirines. *J. Am. Chem. Soc.* **143**, 11337-11344 (2021).

2) Some uncommon abbreviations require detailed explanations, for example, Cu(tfacac)₂ and Cu(hfacac)₂.

Reply: We have added an explanation of the abbreviations, as shown below, in the main text.

Cu(acac)₂ : copper acetylacetonate

Cu(tfacac)₂ : copper trifluoroacetylacetonate

Cu(hfacac)₂ : copper hexafluoroacetylacetonate

3) I am wondering if diazo compounds with different structures could be tolerated, for example, α -diazo malonic esters or α -diazoketones.

A variety of diazo compounds, such as ethyl diazo acetate, vinyl diazo acetate, diazo keto ester, methyl α -diazo malonate, and diazo oxime ether, were examined in the presence of $\text{Cu}(\text{hfacac})_2$ (2.0 mol %) in 1,4-dioxane at 40 °C for 5 h. (Table S2). Ethyl diazo acetate and vinyl diazo acetate was not effective (entries 1 and 2). Diazo keto ester was reacted with azulene to selectively provide alkylation product in 53% yield (keto: enol = 2:1) (entry 3). Methyl α -diazo malonate gave rise to ring expansion product and alkylation product in 39% and 25% yields, respectively (entry 4). When diazo oxime ether was reacted with azulene, the ring expansion product was selectively produced in 83% yield (entry 5). Table S2 was added to the Supporting Information.

Table S2. Examination of Diazo Compounds in Reaction with Azulene in the Presence of $\text{Cu}(\text{hfacac})_2^a$

Entry	Diazo Compounds	Products
1		0
2		0
3		 53% (2:1) ^b
4		 39% 25%
5		 83%

^aAzulene (0.2 mmol, 1.0 equiv) was reacted with diazo compound (1.0 equiv) in the presence of $\text{Cu}(\text{hfacac})_2$ (2.0 mol %) in 1,4-dioxane (0.1 M) at 40 °C for 5 h under a N_2 atmosphere. ^bRatio of keto and enol form.

A number of diazo compounds, such as ethyl diazo acetate, diazo keto ester, methyl α -diazo malonate, diazo oxime ether, diazo Meldrum's acid, and *N*-methyl 3-diazooxindole, were examined in the presence of AgOTf (2.0 mol %) in DCE (0.05 M) at 25 °C for 1 h (Table S3). As can be seen, all of the diazo compounds examined were ineffective. Table S3 was added to the Supporting Information.

Table S3. Examination of Diazo Compounds in Reaction with Azulene in the Presence of AgOTf^a

Entry	Diazo Compounds	Products
1		0
2		0
3 ^b		 trace
4		0
5		0
6		0

^aAzulene (0.2 mmol, 1.0 equiv) was reacted with diazo compound (1.5 equiv) in the presence of AgOTf (2.0 mol %) in DCE (0.05 M) at 25 °C for 1 h under a N₂ atmosphere. ^bReaction was carried out at 70 °C for 12 h.

4) Since the authors proposed that the active catalyst is Cu(I) instead of Cu(II), it's better to add some Cu(I) catalyst entries in the screening table.

CuCl and CuBr as Cu(I) catalysts were examined (Table 1, entries 1 and 2).

Table 1. Reaction Optimization^a

Entry	Cat (mol %)	Solvent	Temp (°C)	Conv (%)	Yield (%) ^b	
					3a	4a
1	CuCl	DCE	25	100	36	7
2	CuBr	DCE	25	80	45	9
3	CuCl ₂	DCE	70	100	0	0
4	CuBr ₂	DCE	70	100	0	0
5	Cu(OAc) ₂	dioxane	70	70	39	7
6	Cu(OAc) ₂ ·H ₂ O	dioxane	70	65	38	9
7	Cu(OTf) ₂	dioxane	25	100	37	0
8	Cu(hfacac) ₃	dioxane	40	62	55	2
9	Cu(hfacac) ₂	dioxane	40	78	47	1
10	Cu(hfacac) ₂	dioxane	40	87	8 ^c (78) ^d	0
11	Cu(hfacac) ₂	hexane	25	84	58	1
12	Cu(hfacac) ₂	toluene	25	100	84	1
13	Cu(hfacac) ₂	DCE	25	100	84 (82) ^d	0
14	Cu(hfacac) ₂	CHCl ₃	25	98	88	2
15	Cu(hfacac) ₂	MeCN	40	85	53	4
16	Cu(hfacac) ₂	DMF	40	30	8	0
17 ^d	Cu(hfacac) ₂	DCE	25	100	95 (93) ^d	0
18 ^{d,e}	Cu(hfacac) ₂	DCE	25	100	90	0
19 ^{d,f}	AgOTf	DCE	25	100	0	74 ^g (5) ^{h,g}

^aAzulene (**1a**, 0.2 mmol, 1.0 equiv), diazo compound (**2a**, 1.0 equiv), and catalyst (2.0 mol %) were used in solvent (1.0 mL, 0.1 M) under a N₂ atmosphere. ^bNMR yield with CH₂Br₂ as an internal standard. ^cIsolated yield. ^dDCE (4.0 mL, 0.05 M) was used. ^eCatalyst (1.0 mol %) was used. ^f**1a** (2.0 equiv) and **2a** (0.2 mmol, 1.0 equiv) were used. ^g1,3-Dialkylated product.

5) In page 11 line 224, enolate is probably not ‘a hard Lewis base’ and this theory may not give convincing explanations on the reactivity difference. A detailed discussion is suggested.

Reply: We thank the referee for the feedback on our use of the HSAB principle to explain the reactivity difference we observed. We are confident that this principle is the most plausible way of explaining our results. The key difference between the two systems discussed in our work is the transition metal used, with copper and silver being isoelectronic but with different properties. Notably, the silver ion is larger than the copper ion, making it more polarizable and softer. Our calculations clearly suggest that this electronic feature is what gives rise to the observed chemoselectivity.

We appreciate the referee's point about the enolate not being considered a hard Lewis base by some researchers. To address this concern, we have added the following statement to the discussion:

"Note that our characterization of the enolate as a relatively hard base should be seen in the context that its calculated interaction with the harder Lewis acid, copper, is stronger than that with the softer Lewis acid, silver. It is not our intention to label the enolate as a hard Lewis base on an absolute scale."

We hope this clarification addresses any confusion and thank you for your valuable feedback on our manuscript.

6) The authors claimed that they conducted ‘energy decomposition analysis’ on the key transition states. In both the manuscript and SI, we didn’t find any reference or procedures about the technique details of the ‘energy decomposition analysis’. According to our understanding, the energy decomposition analysis is to separate an interaction into three parts: electrostatic interaction, repulsive exchange interaction and orbital interaction (see: *WIREs Comput. Mol. Sci.* **2018**, 8: e1345. doi: 10.1002/wcms.1345). What the authors did is more like a ‘distortion-interaction analysis’ (see: *ACIE*, **2017**, 56, 10070). However, what the authors showed in the SI is still very confusing. In Fig. S4, the interaction should decrease the overall barrier instead of increasing it. We strongly suggest to include more details or references for better demonstration.

Reply: The reviewer makes a fair point and we agree that the term 'distortion-interaction analysis' better describes our method of analysis. Although the procedure was initially carried out as part of a more comprehensive energy decomposition analysis, the term 'distortion-interaction analysis' has been widely adopted in the literature and is more appropriate for our work. We have updated our manuscript to use this term consistently.

We also acknowledge the reviewer's concern about our description of the interaction energy in the supplementary information. We apologize for any confusion and would like to clarify that what we denoted as " Δ Interaction" in Figure S4 is the difference in the interaction energy between the transition state and the intermediate, not the interaction energy itself, which is indeed negative. The positive value of Δ Interaction reflects the fact that the interaction energy increases in the transition state due to the reduced electrostatic interaction between the carbanion and metal ion. In the case of copper, the smaller Δ Interaction compared to silver can be attributed to the stronger interaction between the copper ion and carbanion. We hope this clarification addresses the reviewer's concern.

(for reference) Distortion-interaction analysis ↓

Figure S4

7) The authors proposed that both C–H insertion and ring expansion pathways went through the initial addition step with a common intermediate A4 or B4. They said in the SI that the concerted C–H insertion transition states could not be located, but no discussion about the concerted cyclopropanation pathway was

included. I am wondering if a concerted cyclopropanation pathway without the insertion intermediate is possible.

Reply: We appreciate the reviewer's input on the potential for a concerted cyclopropanation mechanism. However, after careful consideration and analysis, we did not deem it a likely pathway for the following reasons. First, the formation of the insertion intermediate is energetically favored, and the intermediate's positive charge is delocalized over several aromatic carbons, making it stable. Additionally, the next cyclopropanation step is also energetically favorable and has a solution-phase free energy for the transition state below 0 kcal/mol in the copper case, indicating a barrierless reaction. Although the silver catalyst favors the C–H insertion pathway, the energy profile still does not support a concerted mechanism. Therefore, we believe that a stepwise mechanism involving the formation of the insertion intermediate is the most plausible explanation for the observed reaction.

(for reference) Concerted C-H insertion pathway (top)

Figure S1. Possible pathways for C–H insertion reaction. Numbers in parenthesis indicate a relative solution phase free energy

8) ¹³C spectra of **1v** and **3v** are misinterpreted. At least three quartet should be observed due to fluorine coupling with coupling constant values of ~ 270 Hz (1J), ~ 30 Hz (2J), ~ 4 Hz (3J). We suggest more scans on a more concentrated sample for better spectra.

After **1v** and **3v** were purely synthesized again, their ¹³C spectra were reinterpreted. Three quartets in ¹³C spectra of **1v** and **3v** were clearly observed due to fluorine coupling with coupling constant values of ~ 270 Hz (1J), ~ 30 Hz (2J), ~ 4 Hz (3J).

1-(4-(Trifluoromethyl)phenyl)azulene (**1v**)

1v

Blue solid; m.p. 45-47 °C; R_f = 0.3 (EtOAc:hexane = 1:50); ^1H NMR (400 MHz, CDCl_3) δ 8.53 (d, J = 9.8 Hz, 1H), 8.39 (d, J = 9.5 Hz, 1H), 8.02 (d, J = 3.9 Hz, 1H), 7.75–7.70 (m, 4H), 7.64 (t, J = 9.8 Hz, 1H), 7.45 (d, J = 3.9 Hz, 1H), 7.212 (t, J = 9.7 Hz, 1H), 7.206 (t, J = 9.8 Hz, 1H); $^{13}\text{C}\{^1\text{H}\}$ NMR (100 MHz, CDCl_3) δ 142.2, 141.3, 138.7, 137.8, 137.3, 135.6, 135.5, 129.9, 129.6, 128.2 (q, J = 32.3 Hz), 125.7 (q, J = 3.8 Hz), 124.7 (q, J = 271.7 Hz), 124.1, 123.9, 117.9; IR (neat) 2967, 1614, 1570, 1429, 1362, 1324, 1163, 1120, 1065, 1015, 845, 741 cm^{-1} ; HRMS (EI) m/z : $[\text{M}^+]$ Calcd for $\text{C}_{17}\text{H}_{11}\text{F}_3$ 272.0813; Found 272.0811.

Methyl 2-phenyl-4-(4-(trifluoromethyl)phenyl)-2H-benzo[7]annulene-2-carboxylate (3v)

3v

The compound **3v** was prepared according to the same procedure as that of the synthesis of **3a**. The crude residue was purified by Silica gel column chromatography to give **3v** (72.1 mg, 85%) as a red solid. m.p. 27–29 °C; R_f = 0.4 (EtOAc:hexane = 1:7); ^1H NMR (400 MHz, C_6D_6) δ 7.41–7.38 (m, 2H), 7.20 (d, J = 8.0 Hz, 2H), 7.03–6.99 (m, 1H), 6.91 (d, J = 8.0 Hz, 2H), 6.31 (d, J = 2.0 Hz, 1H), 5.98 (d, J = 12.0 Hz, 1H), 5.87 (s, 1H), 5.43 (dd, J = 11.1 Hz, 7.4 Hz, 1H), 5.33–5.24 (m, 2H), 3.28 (s, 3H); $^{13}\text{C}\{^1\text{H}\}$ NMR (100 MHz, C_6D_6) δ 173.0, 144.5, 144.3, 139.7, 139.1, 137.2, 136.5, 132.9, 131.5, 129.9, 129.7 (q, J = 32.5 Hz), 129.4, 129.0, 129.0, 127.1, 126.9, 126.4, 125.2 (q, J = 3.7 Hz), 124.3 (q, J = 272.1 Hz), 123.4, 56.4, 52.8; IR (neat) 3058, 3030, 2953, 1733, 1615, 1435, 1325, 1223, 1167, 1124, 1065 cm^{-1} ; HRMS (EI) m/z : $[\text{M}^+]$ Calcd for $\text{C}_{26}\text{H}_{19}\text{F}_3\text{O}_2$ 420.1337; Found 420.1335.

Overall, this is a borderline case. In its current form, this paper is below the standards of *Nat. Commun.*

Reviewer #3 (Remarks to the Author):

In this manuscript, Lee et al. have established the first example of ring expansion and C–H functionalization of azulene with diazo compound catalyzed by transition-metal complexes. The chemoselectivity of the reaction could be controlled simply by changing different transition-metal catalysts. The ring expansion of azulene could be realized by using copper catalyst and the C–H functionalization of azulene could be accomplished using silver catalyst. This strategy has excellent chemoselectivity and functional group tolerance, which could allow the synthesis of a wide range of 6,7-bicyclic compounds and C–H functionalized azulenes. Meanwhile, the reaction mechanism of ring expansion and the C–H functionalization of azulene were studied using DFT method, and the difference in the chemoselectivity of the reaction catalyzed by different metal catalysts was explained. However, the authors have to revise their manuscript to fully address the following issues before a final decision is reached.

1. The authors states that the catalytic reaction begins with CuI(hfacac) as an active catalyst instead of CuII(hfacac)₂, which our calculations show to be ineffective. However, I don't find the related calculations in the manuscript or the supporting materials. Please supply this part or prove the experiment by related copper catalysts, such as CuI(hfacac).

Reply: The referee makes a good point. At the beginning of our studies, we had considered both Cu(I) and Cu(II). However, our computational explorations using Cu(II) afforded many inconsistencies, while Cu(I) consistently gave plausible results. To confirm the oxidation state of the copper catalyst, we carried out several experimental studies, which clearly identified the Cu(I) ion as the catalytically competent species. The comment that the referee refers to is a reflection on our discovery process. Given the experimental evidence, our computational exploration of the Cu(II) reactivity is not necessary for understanding the paper and is not of publication quality. Thus, we removed the reference to the exploratory calculations and have edited the paragraph to read:

“...In the copper catalysis we observed darkening of the reaction mixture by the formation of copper(0) species and the homo-coupling side product of the diazo compound,²⁷ which indicates the formation of copper(I) species...”

2. In the section of Mechanistic studies and DFT calculations, the ring expansion reaction and the C–H functionalization catalyzed by silver catalyst and copper catalyst were calculated, respectively. There is no doubt about the reaction mechanism of azulene catalyzed by silver catalyst, but this is not the case with copper catalyst. Therefore, the authors should discuss not only the difference between TS(X4-X5) and TS(X4-X8) but also the difference in the following fundamental steps in two reaction modes. For the reaction by Ag and Cu catalyst, the authors could use the energetic span model to explain the origin of different chemoselectivity.

Reply: We appreciate the referee's suggestion to use an energetic span model to provide additional analysis. However, we feel that our current approach, which emphasizes the irreversibility of the transformation from A8 to C1 and the relative strengths of the interactions between the catalyst and the carbon, is a simple and intuitive concept that is well-documented through the distortion-interaction analysis. We believe that further computational analysis would not add significant value to this explanation. We will make sure to clarify this point in the manuscript to ensure that readers understand the chemoselectivity of the reaction.

3. In the manuscript, the authors mentioned that the process from A8 to C1 and the process from C1 to C2 are irreversible. Why are they irreversible?

Reply: We want to clarify that we only referred to the reaction from A8 to C1 as being irreversible, not the reaction from C1 to C2. The dissociation of the catalyst from the substrate is an easy and rapid process, and once it occurs, the chance of the catalyst recombining with the intermediate is low due to its dispersion over an infinite solution space at the given concentrations. This gives rise to an irreversible process for the reaction from A8 to C1.

4. In the ring-expansion pathway, is it possible that Cu species didn't dissociate from O to coordinate the EtO-C(=O) moiety? This might make the ring-expansion easier.

Reply: We appreciate the referee's suggestion to investigate the effect of substrate coordination on the chemoselectivity of the reaction. While it is possible that a ring-expansion with coordination of the EtO-C(=O) moiety in the substrate may occur with slightly more preferred energy, we believe that the dissociation of the catalyst is much faster and, therefore, the more dominant process.

5. It is not right to plot a wavy line connecting X5 and X6 in the association step. There is no transition states between them.

Reply: The reviewer is correct that standard quantum chemical calculations cannot locate transition states of simple association reactions. That does not mean, however, that there is no transition state. The free energy barrier of these processes is dominated by the entropy changes, specifically the translational entropy changes. Because standard quantum methods sample the electronic surface only and add the entropy corrections through a simple projection, these transition states on the free energy surface cannot be located precisely. We commented on these issues in unrelated previous work (*JACS* **2002**, *124*, 4495; *Organometallics* **2018**, *37*, 3228). To distinguish transition states that were located vs these estimated ones, we use a dot to identify locations of calculated TS and indicate the TS not explicitly located using the estimated wavy line.

6. The format of G in Fig. 4 should be italic. In addition, the calculated data should keep one decimal place instead of two decimal places, which is beyond the error range of DFT.

Reply: We thank the referee for the suggestion. We changed the format of G in Fig. 4 as the referee suggested.

Determining the appropriate number of significant figures to use in DFT calculations is a complex issue. In our work, we have chosen to display two digits after the comma when using kcal/mol, while in the main text, we round to one digit after the comma. This decision is based on benchmarking DFT methods like B3LYP-D3 with a decent basis set against the G3 database of physical properties, which yields an average error of ~3-5 kcal/mol. This error is considered the "intrinsic system error" of the DFT method, suggesting that these methods are not precise enough to accurately model any chemical reaction of realistic value. For example, a selectivity of 10:1 would require DFT methods to pick up energy differences of ~1 kcal/mol, which would be disqualified by the implied uncertainty of ~3-5 kcal/mol. However, we have found that error-cancellation can significantly benefit DFT methods when comparing very similar molecular systems for related transformations, such as the same catalyst engaging the same substrate to give two different regio- or stereoisomers. In these cases, we estimate the minimum energy resolution to be ~1 kcal/mol, meaning that cautiously examining energy differences of this magnitude may yield meaningful results. It is important to note, however, that this does not automatically mean that a 1 kcal/mol energy difference in DFT can be trusted, as the accuracy of DFT methods can be limited. Nonetheless, our experience suggests that small relative energy differences are often meaningful and worth examining.

Thus, under ideal conditions, we are assuming the energy resolution to be ~1 kcal/mol. It is standard scientific practice to show energies to the last distinct digit, in this case one digit after the comma. That is the reason we use this precision in the main text. And again following good scientific practice, we show one additional digit after the significant one in the raw data, i.e. two digits after the comma. We wish that DFT would be more accurate and we would not have to use such hand-waving arguments to justify the accuracy vs precision, but it is what it is and we are using our best judgement in this case.

REVIEWER COMMENTS

Reviewer #1 (Remarks to the Author):

< In private comments to the Editorial office, Reviewer 2 stated that the concerns of Reviewer 1 from the prior round were adequately addressed. >

Reviewer #2 (Remarks to the Author):

The authors have successfully addressed my previous concerns. The DFT calculation and the related explanation part are now clear and convincing. We are glad that the authors accepted the advice and tried a lot of diazo compounds with diverse structures. I would suggest including the successful examples (Table S2, entry 4 and 5), and briefly discussing about the substrate scope limitations in the main text. The novelty of this work is limited because both pathways are well developed for other electron rich aromatic systems. However, the excellent catalyst-controlled selectivity and the mechanistic insights largely improve the quality of this work.

Reviewer #3 (Remarks to the Author):

The authors didnot directly modify the manuscript according to my concerns like Q3.2, Q3.4, Q3.5 and 3.6 of the third reviewer.

Reviewer #4 (Remarks to the Author):

While the authors' responses to Reviewer #3's points 5 and 6 seem fine to me, I do not find their responses to points 2 and 4 to be satisfactory.

For a catalytic cycle, one should indeed consider energetic span to verify that the proposed selectivity-determining steps do not change.

In addition, modeling reactions of the type described here with Cu can be particularly challenging. Consequently, I think the authors should also do some benchmarking. They comment about the error bars on DFT calculations, but getting reasonable energies for Cu reactions can be quite difficult and it can be quite challenging to assure that a reasonable alternative mechanism was not missed. The functional/basis set combinations chosen here are not generally considered to be among the best for these types of reactions.

Also, the authors argue in some places that catalyst dissociation is easy and irreversible but in other places show steps that appear to require dissociation/rebinding; this may or may not be a contradiction, but we cannot know because "binding energies" appear to be estimated by the authors based only on speculation.

In addition, I am opposed to the authors relegating most computational details to the SI. Readers at least need to be told the exact levels used without having to consult the SI. Readers should not be given the impression that all methods are equal!

The authors might also consider this paper on problems with hard-soft arguments for organic species: "Farewell to the HSAB Treatment of Ambident Reactivity" *Angewandte Chemie International Edition*, Volume 50, Issue 29 p. 6470-6505

The related three references were added.

16. Maeng, C., Son, J.-Y., Lee, S. C., Baek, Y., Um, K., Han, S. H., Ko, G. H., Han, G. U., Lee, K., Lee, K., Lee, P. H. Expansion of Azulenes as Nonbenzenoid Aromatic Compounds for C–H Activation: Rhodium- and Iridium-Catalyzed Oxidative Cyclization of Azulene Carboxylic Acids with Alkynes for the Synthesis of Azulenolactones and Benzoazulenes. *J. Org. Chem.* **85**, 3824-3837 (2020).
17. Maeng, C., Seo, H. J., Jeong, H., Lee, K., Noh, H. C., Lee, P. H. Iridium(III)-Catalyzed Sequential C(2)-Arylation and Intramolecular C–O Bond Formation from Azulenecarboxylic Acids and Diaryliodonium Salts Access to Azulenofuranones. *Org. Lett.* **22**, 7267-7272 (2020).
18. Maeng, C., Lee, P. H. Synthesis of azulenolactones through sequential C(2)-bromoarylation and intramolecular C–O bond formation from azulene-1-carboxylic acids and di(2-bromoaryl)iodonium salts in one pot. *Bull. Korean Chem. Soc.* **43**, 564-569 (2022).

REVIEWER COMMENTS:

Reviewer #2 (Remarks to the Author):

The authors have successfully addressed my previous concerns. The DFT calculation and the related explanation part are now clear and convincing. We are glad that the authors accepted the advice and tried a lot of diazo compounds with diverse structures. I would suggest including the successful examples (Table S2, entry 4 and 5), and briefly discussing about the substrate scope limitations in the main text. The novelty of this work is limited because both pathways are well developed for other electron rich aromatic systems. However, the excellent catalyst-controlled selectivity and the mechanistic insights largely improve the quality of this work.

Reply: In addition, a variety of diazo compounds, including ethyl diazoacetate, ethyl vinyl diazoacetate, α -diazo- β -keto ester, diazomalonate, and α -diazo oxime ether, were examined (see the Supporting Information Table S2). Ethyl diazoacetate and ethyl vinyl diazoacetate was not effective. Although reaction of azulene with diazomalonate provided the ring expansion and C–H alkylation product (39% and 25%, respectively), α -diazo- β -keto ester and α -diazo oxime ether were selectively converted to the C–H alkylation product (53%) and the ring expansion product (83%). These results indicate that the structure of diazo compound is critically important for successful reactions. Other reaction parameters were tested, as discussed in the Supporting Information.

Reviewer #3 (Remarks to the Author):

The authors did not directly modify the manuscript according to my concerns like Q3.2, Q3.4, Q3.5 and 3.6 of the third reviewer.

Reply: We apologize for not having addressed all of this reviewer's comments in a satisfying manner. Also considering Reviewer 4's comments, we have now fully addressed the remaining issues as detailed below.

Reviewer #4 (Remarks to the Author):

While the authors' responses to Reviewer #3's points 5 and 6 seem fine to me, I do not find their responses to points 2 and 4 to be satisfactory.

Reply: In light of the concerns expressed by both reviewers 3 and 4 regarding our previous proposal that the Cu-metal detaches from the substrate, we have undertaken a thorough reevaluation of the proposed mechanism. Our studies confirmed the instincts of these reviewers to be correct, as we have identified a low-energy pathway in which the catalyst remains bound to the substrate. The manuscript has been updated accordingly to reflect these new findings. We wish to express our gratitude to reviewers 3 and 4 for bringing this issue to our attention, ultimately leading to an improved mechanistic proposal.

The figure below highlights the improvement: The previously proposed catalyst dissociation step has been removed simplifying our mechanistic proposal notably. However, our core message remains fully valid and unaltered: the distinct interactions of Cu and Ag with the substrate dictate the chemoselectivity observed.

Previous proposal:

Improved proposal:

For a catalytic cycle, one should indeed consider energetic span to verify that the proposed selectivity-determining steps do not change.

Reply: We believe it is important to emphasize that the primary mission of the computer models employed in this study is to provide a comprehensive understanding of the observed chemoselectivity. With the aim of reaching a broad audience, our intention was to offer a qualitative and plausible explanation supported by detailed calculations, without an overreliance on numerical values. During the previous round of revisions, we held the view that additional computational work was unlikely to significantly enhance the explanatory power of our calculations in this context. However, we duly acknowledge the reviewer's valid perspective regarding the energy span of the entire catalytic cycle, and in response, we have conducted the necessary energy span calculations.

Our findings have confirmed that the fundamental steps governing the selectivity remain unchanged. To clarify our standpoint, we have created a figure depicting an energy profile encompassing the second cycle of the reaction, as illustrated below. Given the substantially lower energy of the C-H alkylation product, the reaction is inherently irreversible, which is why we did not delve into its second cycle. Since silver catalysis predominantly favors the C-H alkylation pathway, our discussion on the energetic span model is focused solely on the ring expansion catalyzed by copper. In this context, the ring-expanded product **A9** serves as the resting state, and the subsequent N_2 extrusion in the second cycle, denoted as **TS(A2-A3)**, incurs an energy cost of 15.6 kcal/mol. This result underscores that the N_2 extrusion in the second cycle does not exert a significant impact on the overall selectivity of the reaction.

In addition, modeling reactions of the type described here with Cu can be particularly challenging. Consequently, I think the authors should also do some benchmarking. They comment about the error bars on DFT calculations, but getting reasonable energies for Cu reactions can be quite difficult and it can be quite challenging to assure that a reasonable alternative mechanism was not missed. The functional/basis set combinations chosen here are not generally considered to be among the best for these types of reactions.

Reply: We concur with the reviewer's assertion that obtaining reasonable energies for reactions of this nature presents a significant challenge. In our pursuit of a suitable computational approach, we systematically compared several well-established functionals with dispersion correction. Our investigations revealed that the Minnesota functional yielded results that not only provided a reasonable explanation for the observed selectivity but also produced energies that align with experimental observations. While B3LYP-D3 yielded nearly identical structures, it exhibited inconsistencies in the relative energies of various intermediates, which did not fully align with experimental data. It is important to note that, as the reviewer correctly implied, silver catalysis posed no significant issues, and we were able to obtain energies that closely matched experimental observations across different functionals.

However, we must acknowledge the presence of erratic and seemingly chaotic behavior in calculations involving various functionals in conjunction with the D3 correction. This phenomenon, although frustrating, is a commonly observed behavior. We believe that some of these issues may stem from errors in the numerical implementations of dispersion corrections rather than being physically meaningful. Nonetheless, given the critical role of dispersion corrections in our study, we believe it is imperative to include them. The Minnesota functionals, by implicitly incorporating short-range dispersion corrections, offer a potential compromise in this regard.

We recognize that the inconsistency and erratic behavior of DFT functionals are challenges that the scientific community grapples with collectively. In our approach, we exercise caution when interpreting computed numbers and focus on identifying generalizable concepts that align with current chemical knowledge. This approach allows us to provide plausible explanations within the context of our study, despite the challenges posed by the numerical intricacies of dispersion corrections and DFT functionals. We believe that our current study effectively demonstrates this cautious approach.

TS(A6-A7)

TS(B6-B7)

		TS(X6-X7)	X6	X4	X8	TS(X8-A9)
b3lyp-d3	Cu	4.00	-5.25	0.00	-1.57	13.37
	Ag	4.58	-5.46	0.00	2.69	17.81
m06	Cu	15.90	5.96	0.00	-7.28	12.79
	Ag	9.98	-1.66	0.00	-3.29	15.48
bp86-d3	Cu	-4.60	-9.19	0.00	-2.05	10.37
	Ag	-3.33	-9.43	0.00	5.48	18.00
pbe0-d3	Cu	4.83	-5.48	0.00	-9.11	9.36
	Ag	4.00	-6.32	0.00	-2.69	15.32
b3pw91-d3	Cu	-0.06	-8.16	0.00	-6.29	10.39
	Ag	-0.42	-9.59	0.00	-0.45	16.34
b1b95-d3	Cu	1.33	-8.16	0.00	-8.30	8.92
	Ag	-2.22	-11.64	0.00	-3.08	15.14
pw6b95-d3	Cu	7.62	-4.28	0.00	-7.54	9.91
	Ag	4.88	-6.62	0.00	-2.41	15.34

Also, the authors argue in some places that catalyst dissociation is easy and irreversible but in other places show steps that appear to require dissociation/rebinding; this may or may not be a contradiction, but we cannot know because "binding energies" appear to be estimated by the authors based only on speculation.

Reply: It appears there may have been a misunderstanding regarding the nature of our discussion in the manuscript. We want to clarify that we were not speculating about binding energies, which can be calculated precisely, but rather speculating about the rates of the associated reaction steps. Quantum chemical calculations provide precise (not necessarily accurate) information about binding energies and barriers, but they cannot estimate the pre-exponential factor of the Arrhenius equation. Therefore, any statement about the rate of a chemical step based solely on calculated barriers inherently involves speculation about the collision factor. This issue becomes particularly complex when comparing barriers for intramolecular rearrangements with those for intermolecular events, as the lack of knowledge about the collision factor can be problematic.

To address this challenge, we exercised judgment in determining whether a reaction step is reversible or irreversible by making plausible assumptions about the concentrations of the involved species at the moment of intermolecular reactions. This approach was necessary because we were previously unable to identify a competitive reaction pathway for the copper-catalyzed cyclization.

Following the reviewer's comments, we reexamined the cyclization step, as mentioned earlier, and were pleased to find a viable cyclization reaction without the need to invoke the detachment of the copper catalyst. This discovery has eliminated the need for speculations about the collision factor, enabling us to streamline the discussion and present a more grounded and precise analysis.

In addition, I am opposed to the authors relegating most computational details to the SI. Readers at least need to be told the exact levels used without having to consult the SI. Readers should not be given the impression that all methods are equal!

Reply: This is a good point. We have now incorporated the computational details into the main part of the manuscript.

The authors might also consider this paper on problems with hard-soft arguments for organic species: "Farewell to the HSAB Treatment of Ambident Reactivity" *Angewandte Chemie International Edition*, Volume 50, Issue 29 p. 6470-6505

Reply: We appreciate the reviewer's insightful comment regarding the limitations of the HSAB concept. Over the course of the past three decades of conducting computational mechanistic studies, we have indeed encountered numerous instances where the HSAB concept does not provide a complete explanation. It is a hallmark of sound scientific practice to question and challenge even well-established textbook knowledge.

That being said, it is important to acknowledge that for every example where the HSAB concept may fall short, there are countless other instances where it remains a valuable and reliable framework. In the context of our current study, we firmly believe that the differential interactions of Ag and Cu with the relatively soft substrate present a plausible and useful explanatory concept. While we remain open to the possibility that alternative explanations may exist for the chemoselectivity observed, we maintain that our proposal is the most reasonable and scientifically justified interpretation based on the evidence at hand.

It is worth noting that, while it would be intriguing to assert that the HSAB concept is fundamentally flawed and in need of a radical revision, our current study does not provide sufficient evidence to support such a dramatic, potentially exciting claim. Instead, we prefer to adhere to a cautious and evidence-based approach in our interpretations, striving to offer the most reasonable and grounded explanations for the observed reaction at hand.

REVIEWERS' COMMENTS

Reviewer #4 (Remarks to the Author):

Overall, I am satisfied with the authors' revisions. I am glad that the reviewers' comments were helpful.

The authors might proofread the new text. It seems to have a variety of grammatical errors.

Also, the authors might consider adding their rationale for using the Minnesota functional to the methods section.

Final comments of Reviewer 4:

The reviewer was satisfied with our revision and requested on final change:

Also, the authors might consider adding their rationale for using the Minnesota functional to the methods section.

Reply:

The energies of the optimized structures were reevaluated with a triple- ζ basis set cc-pVTZ(-f)43 for the main group elemental and with LACV3P for the copper and silver, using the Minnesota functional M0644, as this protocol produced reaction energies that were most consistent with the experimental observations.